# ACTION DIMENSION COORDINATION VIA CENTRALISED CRITICS FOR CONTINUOUS CONTROL

## ABSTRACT

Continuous control tasks with large action spaces often demand coordination across action dimensions. Recent work has shown that factorising the action space enables deep Q-learning to tackle high-dimensional continuous control problems by leveraging value decomposition methods adapted from multi-agent reinforcement learning (MARL). However, these approaches treat action dimensions independently, which can result in sub-optimal policies when coordination is required. To overcome this, we propose a general framework that adapts centralised training with decentralised execution (CTDE) to single-agent continuous control with factorised action spaces. Our key insight is to reinterpret action dimensions as cooperative "agents" and enable them to exchange information via a centralised critic during training, leading to coordinated policies that can be executed in a decentralised manner at test time. We instantiate this framework with two algorithms, DAC-AC and DAC-DDPG, and evaluate them on 13 DeepMind Control Suite tasks, demonstrating that incorporating centralised critics improves both sample efficiency and asymptotic performance on a wide range of tasks. Using these two algorithms, we further show that our framework seamlessly integrates with existing offline RL methods, achieving state-of-the-art performance across multiple benchmarks.

## 1 INTRODUCTION

Reinforcement learning (RL) has emerged as a popular framework for learning optimal control without requiring prior knowledge of model dynamics. Notable breakthroughs include applications to large language models (LLMs) (Guo et al., 2025; Havrilla et al., 2024), autonomous driving (Shi et al., 2021; Li et al., 2023), and game playing (Mnih et al., 2015). Despite this, scaling RL to complex continuous control remains challenging, particularly when action spaces are high-dimensional and require careful coordination between action dimensions. Standard actor-critic methods often struggle in such settings due to poor sample efficiency and optimisation instability.

Recently a growing body of research has emerged that addresses scalability in RL algorithms by factorising continuous actions and applying value decomposition methods from multi-agent RL (MARL) (Seyde et al., 2023; Tang et al., 2022; Ireland & Montana, 2024). Traditional discrete-action methods, such as DQN (Mnih et al., 2015), struggle when faced with higher-dimensional action spaces due to the exponential increase in the number of Q values that must be learnt. By treating action dimensions independently, these approaches reduce the exponential cost of learning joint Q-values to linear in the number of dimensions. However, this independence assumption fails when action dimensions are interdependent, often resulting in suboptimal policies in high-dimensional control tasks.

In contrast to value decomposition methods, the MARL community has developed coordination mechanisms, most notably the centralised training with decentralised execution (CTDE) framework, that explicitly shares information across agents during training. Algorithms such as MAAC (Iqbal & Sha, 2019) and MADDPG (Lowe et al., 2017) have shown that centralised critics significantly improve sample efficiency and policy quality by enabling coordinated behaviour.

Our key insight is that action dimensions in factorised action spaces are analogous to agents in MARL: they often interact and must be coordinated to achieve high performance. We therefore adapt CTDE methods to single-agent settings by equipping decomposed critics with centralised

conditioning on action dimensions. This allows each dimension to learn not in isolation, but in the context of others.

Our contributions are laid out as follows:

- We introduce a general CTDE-inspired framework for coordinating action dimensions in continuous control
- We propose two concrete algorithms: DAC-AC, an actor-critic algorithm modelled on MAAC, and DAC-DDPG, an actor-critic method modelled on MADDPG.
- We demonstrate consistent improvements across 13 DeepMind Control Suite tasks, with strong gains in high-dimensional environments (Humanoid, Dog).
- Using these two algorithms, we show that our framework integrates seamlessly with existing offline RL methods, achieving state-of-the-art results on multiple benchmarks.

By bridging MARL coordination mechanisms and single-agent RL, our work highlights the importance of action-dimension coordination and establishes a foundation for further exploration of cross-dimensional dependencies in continuous control.

## 2 BACKGROUND

**Single agent value decomposition**    Seyde et al. (2023) address the challenge of solving continuous state-action problems using algorithms originally designed for discrete action spaces. They factorise the action space using a bang-off-bang procedure and introduce Decoupled Q-Networks (DecQN), which combines deep Q-learning (Mnih et al., 2015) with VDNs (Sunehag et al., 2018), a concept from MARL, to learn separate utility values for each action dimension. By treating each action dimension independently, their method reduces the computational complexity from exponential to linear in the number of action dimensions.

Ireland & Montana (2024) extend this work by leveraging ensemble methods to reduce variance and improve sample efficiency. Their algorithm, REValueD, also introduces a regularisation term to address the credit assignment issue in DecQN. Through our experiments, we demonstrate that this regularisation term has significantly less impact than the use of ensembles in improving performance. In contrast, the centralised critic architecture we use shows improvement in both convergence rate and asymptotic performance across a range of tasks. Other algorithms alternatively explore adaptively adjusting the level of discretisation to enable finer control in complex environments (Seyde et al., 2024; Seo et al., 2025).

Lee et al. (2025) use a model-based approach for handling factorised action spaces. While their method shares a similar decomposed architecture to DecQN, it additionally learns a forward dynamics model and a reward model for each action dimension. Although this approach improves sample efficiency in settings with low state-action dimensionality, its asymptotic performance tends to underperform compared to DecQN and it faces significant limitations in more complex environments. In particular, state reachability issues associated with training the forward dynamics model (Edwards et al., 2020; Hepburn et al., 2024) have been shown to lead to learning failure as the complexity of the environment increases.

**Cooperative multi-agent RL**    Traditional single agent RL algorithms do not scale effectively to multi-agent environments, as simply extending them by concatenating all agents' observations and actions into a single joint input results in an exponentially growing state-action space that severely impacts learning efficiency. At the other extreme, methods that treat agents as fully independent learners struggle in scenarios that require coordination and collaboration (Tan, 1993). To address these issues, the centralised training with decentralised execution (CTDE) framework has been introduced in MARL. CTDE enables agents to share information and coordinate during training while ensuring that policies can be executed independently, without communication, at test time.

MAAC (Iqbal & Sha, 2019) is one such algorithm that builds upon the CTDE framework, extending soft actor-critic (SAC) (Haarnoja et al., 2018) by incorporating an attention mechanism (Vaswani et al., 2017) into the critic. This attention-based critic selectively conditions on observations and actions from other agents, enabling more efficient information sharing during training.

Similarly, MADDPG (Lowe et al., 2017) extends Deep Deterministic Policy Gradient (DDPG) (Silver et al., 2014) to the multi-agent setting by employing a centralised critic that explicitly conditions on all agents' actions during training. In contrast, VDNs assume conditional independence between agents, factorising the global Q-value into individual agent contributions without explicit information sharing.

The effectiveness of these CTDE-based coordination mechanisms over independence assumptions has been consistently demonstrated in empirical studies. Benchmark studies illustrate these performance differences clearly. Bettini et al. (2024) develop a standardised benchmark library demonstrating MADDPG outperforming VDN. Additionally, Utke et al. (2025) show that MAAC achieves superior performance compared to QMIX, an advanced, more generalised extension of VDN, across multiple cooperative environments.

**Offline RL** Offline RL focuses on training agents using pre-collected datasets, thereby avoiding the need for potentially costly or dangerous online interactions. A central challenge in offline RL is addressing the distributional shift that arises when the agent overestimates the value of state-action pairs that lie outside of the support of the dataset (i.e., out-of-distribution or OOD). Most offline methods address this challenge through one of two strategies: policy constraint or conservative value estimation. Policy constraint methods (Fujimoto & Gu, 2021; Fujimoto et al., 2019) mitigate distributional shift by explicitly regularising the learned policy to remain close to the behaviour policy that generated the dataset. In contrast, conservative value estimation methods (An et al., 2021; Kumar et al., 2020; Wu et al., 2019) implicitly regularise the agent's policy by penalising Q-value estimates for OOD state-action pairs.

Beeson et al. (2024) introduce a suite of offline benchmarks designed to evaluate discrete-action algorithms in continuous-action environments. They apply existing policy constraint and conservative value estimation methods to regularise DecQN and demonstrate its effectiveness in learning from a fixed offline dataset. While many prior approaches are built around actor-critic architectures, DecQN is a value-based algorithm. As a result, additional adaptations are necessary to make these regularisation techniques compatible with DecQN. In contrast, our methods based on the CTDE framework naturally incorporate actor-critic architectures, allowing us to directly use existing offline RL methods to prevent distributional shift without requiring specialised adaptations.

## 3 PRELIMINARIES

We formulate the RL problem as a Markov decision process (MDP) defined by the tuple $(\mathcal{S}, \mathcal{A}, P, r, \rho, \gamma)$, where $\mathcal{S} \in \mathbb{R}^{d_s}$ and $\mathcal{A} \in \mathbb{R}^N$ denote the state and action spaces, $P(\mathbf{s}'|\mathbf{s}, \mathbf{a})$ specifies the environment dynamics, $\rho(\mathbf{s}_0)$ is the initial state distribution, $r : \mathcal{S} \times \mathcal{A} \to \mathbb{R}$ is the reward function, and $\gamma \in (0, 1]$ is the discount factor. The agent's behaviour is governed by a policy $\pi(\mathbf{a}|\mathbf{s})$ and the expected return for a given policy can be expressed using the action value function $Q^\pi(\mathbf{s}, \mathbf{a}) = E_\pi[\sum_{t \geq 0} \gamma^t r_t | s_0 = \mathbf{s}, a_0 = \mathbf{a}]$. The goal in RL is to find an optimal policy $\pi^\star$ that maximises the expected discounted return.

$$\pi^\star = \arg\max_\pi J(\pi) := E_{s \sim \rho, a \sim \pi(\cdot|s)} \left[ Q^\pi(\mathbf{s}, \mathbf{a}) \right].$$

Ireland & Montana (2024) describe an action space as being factorisable if $\mathcal{A}$ can be decomposed into a set of discrete sub-action spaces $\mathcal{A}_1 \times \cdots \times \mathcal{A}_M$. Like both Ireland & Montana (2024) and Seyde et al. (2023), we use bang-off-bang discretisation, and hence focus on the case where $M = N$.

## 4 GENERAL CTDE FRAMEWORK FOR ACTION DIMENSIONS

We propose a general framework that brings centralised training with decentralised execution (CTDE) from MARL to single-agent continuous control with factorised actions. The key idea is to treat each action dimension as a cooperative "agent" during training and let its critic condition on a subset of the action dimensions.

Concretely, let the joint action be $\mathbf{a} = (a_1, \ldots, a_N) \in \mathbb{R}^N$. For each dimension $i$, choose an index set

$$\mathcal{C}_i \subseteq \{1, \ldots, N\}$$

that specifies which action dimensions constitute the conditioning context for $i$. We then learn a centralised decomposed critic value

$$Q_\theta^i(\mathbf{s}, \mathbf{a}_{\mathcal{C}_i})[a_i]$$

for each discrete value of $a_i$ conditioned on the state $\mathbf{s}$ and contextual subset $\mathbf{a}_{\mathcal{C}_i}$. The notation $Q(\mathbf{s}, \mathbf{a}_{\mathcal{C}_i})[\cdot]$ highlights that the critic value is selected by using $a_i$ to index the output of the architecture. Unlike Ireland & Montana (2024), we refer to these decomposed functions as critics instead of utility functions to emphasise their action-conditioned inputs.

During training, critics are centralised because they share action information as described by $\mathbf{a}_{\mathcal{C}_i}$. At test time, we execute a single actor $\pi_\phi$ that maps $\mathbf{s}$ to the joint action, without access to adjacent action dimensions, so execution remains decentralised.

Examples of $\mathcal{C}_i$:

- $\mathcal{C}_i = \varnothing$ recovers independent utilities (VDN-style)

- $\mathcal{C}_i = \{1, \ldots, N\}$ gives fully centralised conditioning.

- Local/structured context: e.g. $k$-nearest dimensions around $i$

- Learned context (attention)

**Generic training template**   We provide a simple training template outlining how algorithms that fall under this framework are updated.

---

**Algorithm 1** General framework training (1-step; single critic)

---

1: Sample $(\mathbf{s}, \mathbf{a}, r, \mathbf{s}')$ from replay buffer and sample next step action $\mathbf{a}' \sim \pi_\phi(a|s')$
2: For each dimension $i \in \{1, \ldots, N\}$ form the target

$$y_i = r + \gamma \max_{a_i} Q_\theta^i(\mathbf{s}', \mathbf{a}'_{\mathcal{C}_i})[a_i]$$

3: Using the sampled action $\mathbf{a}'$, update each critic by minimising

$$\mathcal{L}_{\text{critic}}(\theta) = L(y_i - Q_\theta^i(\mathbf{s}, \mathbf{a}_{\mathcal{C}_i})[a_i]),$$

where $L(\cdot)$ is the Huber loss.
4: Update $\pi_\phi(\cdot|s)$ so it outputs actions whose per-dimension choices align with the critics, either via imitation of the critics' greedy actions, or via a policy gradient surrogate objective.

---

This template decouples (i) the decomposed-critic backbone and choice of $\mathcal{C}_i$ from (ii) the actor update rule, enabling multiple instantiations under a unified design and letting practitioners trade off coordination strength, computation, and sample efficiency by how $\mathcal{C}_i$ is specified or learned.

## 4.1 DECOUPLED Q-NETWORKS

Both Seyde et al. (2023) and Tang et al. (2022) propose the idea of decomposing the Q function, used in Deep Q-Networks (DQN) (Mnih et al., 2015), into utility functions $U^i(\mathbf{s})[a_i]$ for each action dimension. By treating each action dimension independently, the approach reduces computational complexity from exponential to linear in the number of actions.

We can consider this a special case of our framework with $\mathcal{C}_i = \varnothing$ and $U_{\theta_i}^i(\mathbf{s})[a_i] = Q_{\theta_i}^i(\mathbf{s}, \varnothing)[a_i]$. This highlights that DecQN ignores cross-dimensional dependencies, which can lead to suboptimal coordination when actions are coupled in dynamics or reward.

As the individual utility functions do not rely on information from adjacent action dimensions they are effectively decentralised and can be used directly at execution to compute actions. We define the joint action that maximises the utility functions as follows

$$\mathbf{a}^{\text{opt}} := \left( \arg\max_{a_i} U_{\theta_i}^i(\mathbf{s})[a_i] \right)_i.$$

Instead of updating each utility function independently, a global critic value is computed as the average over the utility functions as follows:

$$Q_\theta(\mathbf{s})[\mathbf{a}] = \frac{1}{N}\sum_{j=1}^{N} U_{\theta_j}^{j}(\mathbf{s})[a_j], \tag{1}$$

The Global Q function is then trained by minimising the following loss function

$$\mathcal{L}(\theta) = \frac{1}{|B|}\sum_{(s_0,\mathbf{a}_0,r_{0:n-1},s_n)\in B} L(y^n - Q_\theta(\mathbf{s}_0)[\mathbf{a}_0]) \tag{2}$$

where $B$ is the sampled batch and $y^n = \sum_{j=0}^{n-1}\gamma^j r_j + \gamma^n Q_{\bar{\theta}}(\mathbf{s}_n)[\mathbf{a}^{\text{opt}}]$ is the $n$-step Q-learning target, and $\bar{\theta}$ the lagged parameter of the target Q network.

Ireland & Montana (2024) show that the target variance in DecQN can be reduced with ensembles by replacing $U^j(\mathbf{s})[a_j]$ with the mean $\frac{1}{K}\sum_{k=1}^{K} U_{\theta_{j,k}}^{j,k}(\mathbf{s})[a_j]$ across $K$ ensemble members.

For clarity, Algorithm 1 presents our template for the 1-step variant with a single critic (ensemble size 1) but extending to n-step targets and to ensembles of size $K$ is straightforward.

## 4.2 DAC-AC

Within our framework we propose, Discrete-Action-Conditioned-AC (DAC-AC), an actor-critic instantiation inspired by the multi-agent algorithm MAAC (Iqbal & Sha, 2019). In MAAC, each agent has its own observation and passes $(o_i, a_i)$ through an agent-specific embedding before attention combines information across agents. In our setting, all action dimensions share the same observation, so the attention role reduces to learning how much each dimension's action should influence the others. Because the shared signal is one-dimensional (a single action component), explicit per-agent embeddings are unnecessary.

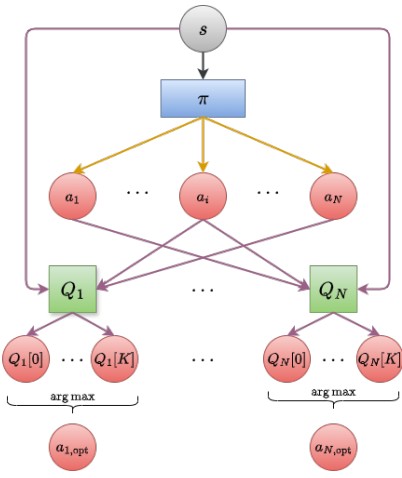

Figure 1: Overview of the interaction between actor and centralised critic in DAC-AC.

**Architecture** The actor $\pi_\phi(a|s)$ factorises across dimensions; head $i$ outputs a categorical distribution over the discretised values $a_i$. The critic for dimension $i$, conditions on the action $\mathbf{a}_{\mathcal{C}_i}$, specified by the index set $\mathcal{C}_i = \{1,\dots,N\}\setminus\{i\}$, which we denote by $\mathbf{a}_{-i}$.

**Critic Update** Using the actor for bootstrapping, draw $a' \sim \pi_\phi(\cdot|s)$ and, for each $i$, compute $y_i = r + \gamma \max_{a_i} Q_{\bar{\theta}_i}^{i}(\mathbf{s}',\mathbf{a}'_{-i})[a_i]$ and update the critic to minimise

$$\mathcal{L}_{\text{critic}}(\theta_i) = L(y_i - Q_{\theta_i}^{i}(\mathbf{s},\mathbf{a}_{-i})[a_i])$$

**Actor Update** Given state $s$, draw $\mathbf{a} \sim \pi_\phi(\cdot|s)$ and compute per-dimension greedy labels

$$a_i^{\text{opt}} := \arg\max_{a_i} Q_{\theta_i}^{i}(\mathbf{s},\mathbf{a}_{-i})[a_i]$$

Update the actor with a cross-entropy loss that matches each policy head to the class of $a_i^{\text{opt}}$. This keeps training centralised, while execution remains decentralised: at test time we take greedy action from $\pi_\phi(\cdot|s)$ for each dimension.

For clarity, we provide a schematic illustrating the interaction between the actor and centralised critic in Figure 1.

### 4.3 DAC-DDPG

As a second instantiation of our framework, we introduce Discrete-Action-Conditioned-DDPG (DAC-DDPG) a policy-gradient variant inspired by MADDPG (Lowe et al., 2017).

**Architecture**   For the actor, each head outputs logits for each possible value of $a_i$, and during training, we obtain differentiable discrete samples with a Gumbel-Softmax transformation. In the multi-agent setting, MADDPG indiscriminately shares all information between agents via each agent's centralised critic. As a result, we condition dimension $i$'s critic on all actions, i.e. $\mathcal{C}_i = \{1, \ldots, N\}$ that we denote $\mathbf{a}$.

**Critic update**   As in Section 4.2, we generate an action using our policy and update the parameters of the critic $Q^i_{\theta_i}(\mathbf{s}, \mathbf{a})[a_i]$ by minimising the loss to the target $y_i$.

**Actor update**   We maximise the critics' values under the current policy using the objective described in DDPG. We first sample $\mathbf{a}^{\mathrm{samp}} \sim \pi(\cdot|s)$ using Gumbel-Softmax and update the actor using the objective

$$L_\pi(\phi) = \sum_{i=1}^{M} Q^i_{\theta_i}(\mathbf{s}, \mathbf{a}^{\mathrm{samp}})[a_i^{\mathrm{samp}}]$$

At evaluation, we take per-dimension $\arg\max$ over logits (decentralised execution).

### 4.4 OFFLINE LEARNING

In addition to the online learning component, our framework can be directly adapted to the offline RL setting. Since our methods adopt actor-critic architectures, they are naturally compatible with policy constraint and conservative value estimation techniques commonly used in offline settings.

To demonstrate this, we implement the behaviour cloning (BC) regularisation approach of (Fujimoto & Gu, 2021), which introduces an auxiliary loss that encourages the actor to produce actions close to those observed in the dataset.

$$L_\pi^{\mathrm{offline}}(\phi) = L_\pi^{\mathrm{online}}(\phi) + \lambda CE(\pi_\phi(\cdot|s), \mathbf{a}_\mathcal{D})$$

where $\mathbf{a}_\mathcal{D}$ is an action sampled from the dataset $\mathcal{D}$ and CE denotes cross-entropy. This formulation helps mitigate overestimation bias by preventing the actor from drifting towards out-of-distribution (OOD) actions, while leaving the critic learning process unchanged. We denote the offline variants of our algorithms as DAC-AC-BC and DAC-DDPG-BC. While we focus on BC regularisation in our experiments, other policy constraint methods could also be integrated into our framework without requiring further structural modifications.

### 4.5 COMPLEXITY

The centralised critic architecture results in each per-dimension critic taking $[\mathbf{s}, \mathbf{a}_{\mathcal{C}_i}] \in \mathbb{R}^{d_s + |\mathcal{C}_i|}$ as input. For an MLP critic, this results in a per-update compute across all $N$ dimensions and ensemble size $K$ scales linearly with $\mathcal{C}_i$ as follows

$$\mathcal{O}(KN(d_s + |\mathcal{C}_i|)H_Q)$$

where $H_Q$ represents the number of dimensions in the hidden layer. We show empirically in Table 2 that this yields a modest computational overhead vs. REValueD on Cheetah-Run (DAC-DDPG $\approx 1.1\times$ and DAC-AC $\approx 1.4\times$).

## 5 EXPERIMENTAL RESULTS

### 5.1 ONLINE RESULTS

In this section we investigate the performance of our algorithms using several benchmarks from the DeepMind Control Suite (Tassa et al., 2018) built upon the MuJoCo (Todorov et al., 2012) physics

engine. We benchmark our method across 13 different environments in the suite, including the Dog environment that has 223 state dimensions and 38 action dimensions. Results for Walker-Run can be found in Appendix A and experimental details are listed in Appendix B.

**Online baselines**   We compare our methods against several established baselines including TD3 (Fujimoto et al., 2018), SAC (Haarnoja et al., 2018), DecQN_N, a variant of the original DecQN (Seyde et al., 2023) algorithm using an ensemble of $N = 10$ critics. Additionally, we evaluate our algorithms against REValueD (Ireland & Montana, 2024), an extension of DecQN_N that incorporates regularisation to explicitly address the credit assignment issue.

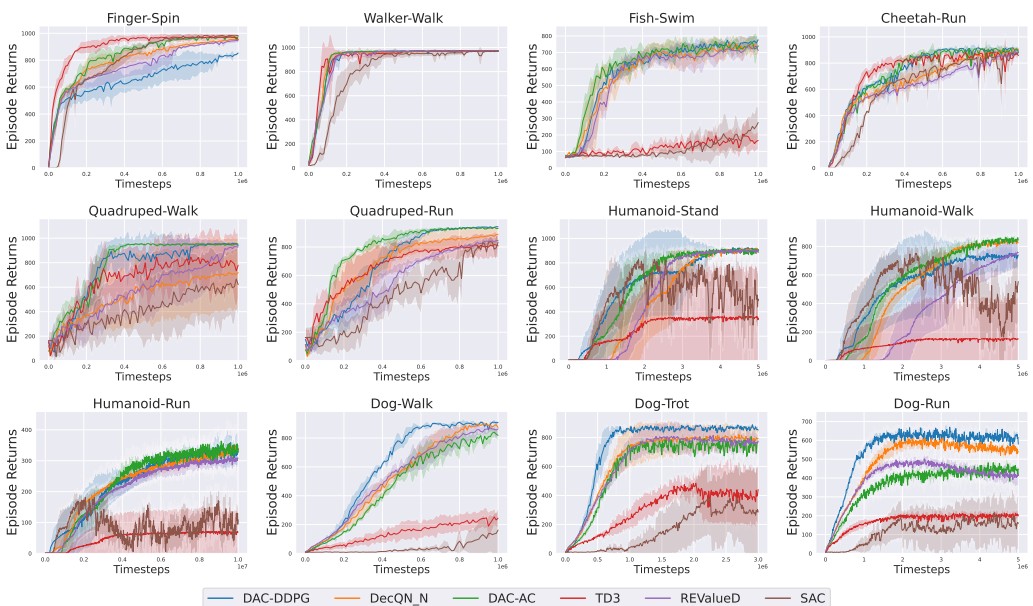

Figure 2: Returns for DeepMind Control Suite tasks. Solid line represents mean over 5 seeds with shaded area corresponding to $\pm 1$ standard deviation. Evaluation is conducted every 10,000 timesteps, and returns are averaged over 10 episodes.

Figure 2 shows that the continuous actor-critic method TD3 excels in tasks with low action dimensionality, but its performance deteriorates significantly in more complex settings such as the Humanoid and Dog environments. Although SAC achieves somewhat competitive performance on the humanoid environments, training is unstable and it performs poorly on most other tasks. This highlights the advantage of discretising the action space and employing a decomposed critic architecture, which tends to scale more effectively in high-dimensional, complex environments.

While Ireland & Montana (2024) demonstrate that their method, REValueD, consistently outperforms DecQN, we find that simply increasing the ensemble size of DecQN yields comparable or better performance to REValueD. This suggests the regularisation term introduced in REValueD to address the credit assignment problem has a limited impact on overall learning performance.

In contrast, both DAC-AC and DAC-DDPG show consistent improvement over both REValueD and DecQN_N across a range of tasks in both convergence speed and asymptotic performance. This underlines the effectiveness of using centralised critic architectures in the online setting.

## 5.2 OFFLINE RESULTS

Beeson et al. (2024) introduce a standardised collection of datasets designed to evaluate offline discrete-action methods in continuous control tasks. Further details are provided in Appendix B.2.2. We refer to the offline variant of our algorithms as DAC-DDPG-BC and DAC-AC-BC.

**Offline Baselines** We compare our algorithm to DecQN-CQL and DecQN-IQL, the two best performing methods introduced by Beeson et al. (2024), as well as behavioural cloning (BC).

Table 1 shows that DAC-AC-BC matches or exceeds the best-performing baselines across all tasks in the random-medium-expert and medium datasets. Furthermore, it outperforms all baselines across all dataset qualities for the Cheetah-Run tasks. This indicates that conditioning the critics' actions can result in better trajectory optimisation when learning from static datasets.

In contrast, DAC-DDPG-BC struggles to match state-of-the-art performance on a wide range of tasks, which we attribute to the bias introduced by the categorical reparameterisation used in the Gumbel-Softmax. We avoid using a straight-through (ST) estimator as it would require one-hot representations, substantially inflating the critic's input dimensionality in high-dimensional spaces. Moreover, ST trades variance for bias, meaning the surrogate gradients remain biased, so it does not address the reparameterisation bias we observe. Additionally, both algorithms struggle on the medium-expert datasets for the Humanoid-Stand and Dog-Trot tasks, likely due to the high action dimensionality of these tasks, to which our algorithms appear more sensitive in the offline setting.

Table 1: Normalised average returns on Factorised action tasks. Scores are averaged across 5 seeds with 10 episodes per seed. Where relevant, we report the mean $\pm$ standard error.

| Dataset | BC | DecQN-CQL | DecQN-IQL | DAC-DDPG-BC | DAC-AC-BC |
|---|---|---|---|---|---|
| Cheetah-Run | | | | | |
| -random-medium-expert | 41.5 | **79.6** | 61.4 | $43.2 \pm 1.31$ | $\mathbf{79.6 \pm 1.89}$ |
| -medium | 40.4 | 48.3 | 47.7 | $42.2 \pm 0.65$ | $\mathbf{53.3 \pm 0.46}$ |
| -medium-expert | 61.6 | 103.2 | 102.5 | $51.8 \pm 4$ | $\mathbf{106.9 \pm 1.65}$ |
| -expert | 99.9 | 105.6 | 104.6 | $99.4 \pm 1.58$ | $\mathbf{106 \pm 0.5}$ |
| Quadruped-Walk | | | | | |
| -random-medium-expert | 28 | 78.7 | 65.8 | $49.3 \pm 6.52$ | $\mathbf{83.5 \pm 10.86}$ |
| -medium | 39.2 | 48.6 | 46.3 | $43.5 \pm 9.37$ | $\mathbf{52.5 \pm 8.43}$ |
| -medium-expert | 63.4 | 115.4 | **121.2** | $70.7 \pm 5.41$ | $115.7 \pm 2.53$ |
| -expert | 97.7 | 118.2 | **122.3** | $96.3 \pm 4.71$ | $120 \pm 2.42$ |
| Humanoid-Stand | | | | | |
| -random-medium-expert | 34.4 | 42.7 | 46 | $41 \pm 1.28$ | $\mathbf{55.3 \pm 2.2}$ |
| -medium | 44.4 | 51.4 | 53.8 | $45.2 \pm 0.39$ | $\mathbf{54.2 \pm 0.96}$ |
| -medium-expert | 63.1 | 104.7 | **113.3** | $58 \pm 1.79$ | $89.5 \pm 1.08$ |
| -expert | 102.2 | 109 | **116.6** | $97.8 \pm 3.23$ | $103.8 \pm 0.75$ |
| Dog-Trot | | | | | |
| -random-medium-expert | 37.2 | 43.4 | 44.1 | $31.6 \pm 3.64$ | $\mathbf{45 \pm 0.27}$ |
| -medium | 43.8 | 46.5 | 52 | $44 \pm 0.59$ | $\mathbf{56.4 \pm 0.58}$ |
| -medium-expert | 62 | 84.8 | **89.3** | $61 \pm 1.54$ | $63.7 \pm 2.17$ |
| -expert | 98 | **99.5** | 98.9 | $92.7 \pm 0.52$ | $93.9 \pm 1.2$ |

## 6 ABLATION STUDIES

**Layer Normalisation** Like both Ireland & Montana (2024) and Seyde et al. (2023), we incorporate layer normalisation (Ba et al., 2016) in our critic architecture, as it has been shown to improve learning performance. Figure 3 reveals, however, that layer normalisation is a critical component in both DecQN and REValueD, necessary to effectively prevent overestimation bias in complex environments such as the Dog tasks.

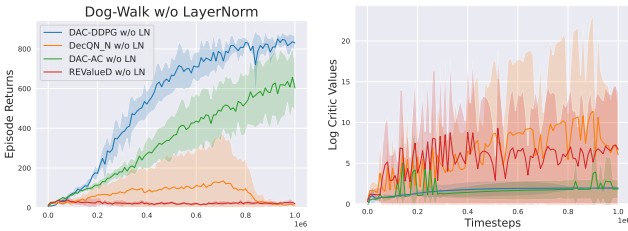

Figure 3: Impact of removing layer normalisation (LN) on algorithm performance in the Dog-Walk task. The left plot shows episode returns, and the right shows the log critic values. Unlike DecQN_N and REValueD, our methods maintain stable learning performance and critic values, demonstrating greater robustness to architectural ablations.

While techniques such as layer normalisation can enhance performance, relying on such components to directly prevent overestimation can reduce interpretability, creating algorithms that are difficult to build upon as task complexity increases.

**Coordination strength via $\mathcal{C}_i$** Figure 4 shows online returns and learned critic values for Quadruped-Walk/Run and Dog-Walk with DAC-AC under varying conditioning set sizes $\mathcal{C}_i$. Across all tasks, larger $\mathcal{C}_i$ leads to faster learning, higher final returns, and higher critic estimates, demonstrating the benefit of incorporating additional action context during training. Notably, performance gains saturate beyond a moderate context size: for Dog-Walk, conditioning on 15 and 30 dimensions yields comparable returns and critic values. This suggests that full centralisation may not be necessary and motivates future work on learning or adaptively selecting $\mathcal{C}_i$ to balance coordination benefits against computational overhead.

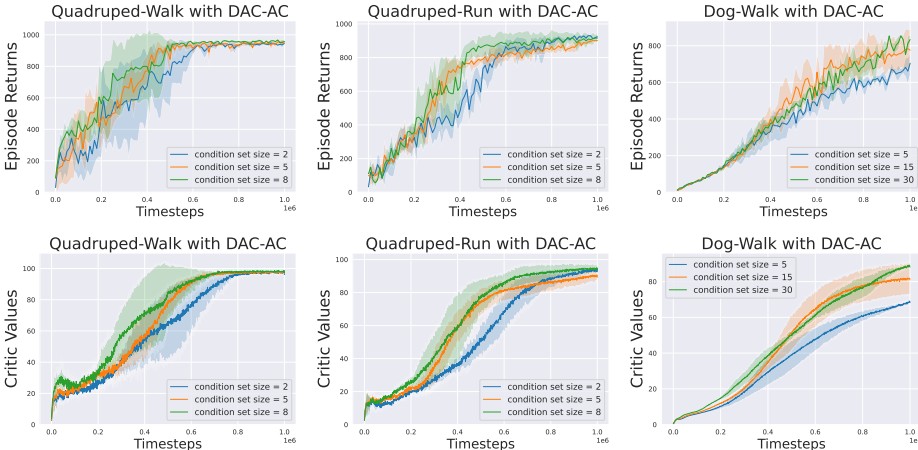

Figure 4: Performance of DAC-AC with varying $|\mathcal{C}_i|$. Top row: episode returns. Bottom row: learned critic values. Solid lines show the mean over 3 seeds, and shaded regions indicate $\pm$ standard deviation. Returns are evaluated every 10,000 timesteps and averaged over 10 episodes.

In Appendix C we vary the actor-update frequency in DAC-AC to assess whether fixing the actor for extended periods yields more stable critic training.

## 7 DISCUSSION AND CONCLUSIONS

This work introduces a unified CTDE framework for coordinating action dimensions in high-dimensional continuous control. By framing action dimensions as cooperative "agents", we enable centralised critics to share context across dimensions during training, improving both credit assignment and coordination quality while maintaining decentralised execution.

Our empirical study shows that this design consistently outperforms prior factorised methods across 13 DeepMind Control Suite tasks with particularly strong gains in high-dimensional action spaces where coordination between action dimensions is most critical. Importantly, the framework's actor-critic foundation allows seamless integration with policy constraint and conservative value estimation techniques from offline RL, achieving state-of-the-art results on multiple offline benchmarks.

However, our work also reveals some limitations that warrant further investigation. Our methods show sensitivity to action dimensionality in offline settings, with reduced performance on some high-dimensional tasks. Additionally, the reliance on bang-off-bang discretisation also introduces approximation errors that could impact performance in tasks requiring fine-grained control. One promising direction for future research is to explore integrating adaptive discretisation schemes such as Seyde et al. (2024) that could broaden the framework's applicability to more complex problems.

Overall, we believe our framework provides a blueprint for designing and evaluating coordination mechanisms in single-agent RL, bridging the gap between MARL and continuous control, and paving the way for more scalable and coordination-aware RL algorithms.

# 8 REPRODUCIBILITY STATEMENT

We provide code as supplementary material to enable reproduction of all experiments presented in this paper. Detailed hyperparameter settings are provided in Appendix B.

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

# A    ONLINE RESULTS

Online results for Walker-Run.

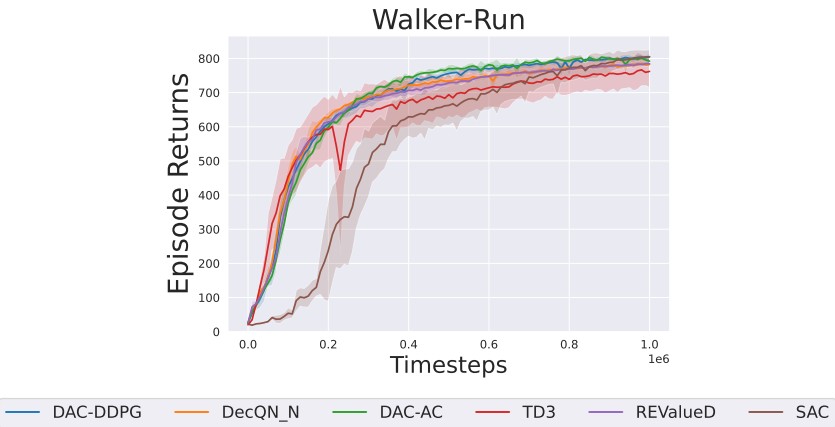

Figure 5: Returns for the Walker-Run task. Solid line represents mean over 5 seeds with shaded area corresponding to $\pm 1$ standard deviation. Evaluation is conducted every 10,000 timesteps, and returns are averaged over 10 episodes.

# B    EXPERIMENTAL DETAILS

## B.1    HARDWARE AND SOFTWARE SPECIFICATIONS

**Hardware**    We run all tests on a single A100 GPU and provide the details of computational cost for training each of the algorithms of our method on Cheetah-Run in Table 2 and Humanoid-Stand in Table 3. We also provide the computational cost of TD3, DecQN_N and REValueD as a comparison and specify the ensemble size of critics used for each algorithm in brackets. While we can reduce DAC-DDPG's memory footprint through vectorisation, DAC-AC's asymmetric conditioning across action dimensions limits how far vectorisation can be applied. If all dimensions instead shared the same conditioning set $\mathcal{C}$, DAC-AC's GPU memory usage would be comparable to DAC-DDPG.

Table 2: Computational cost (Cheetah-Run)

| Method | Runtime (s/epoch$^\star$) | GPU Mem. (MiB) |
|---|---|---|
| TD3 (N=2) | 48.8 | 540 |
| SAC (N=2) | 48.8 | 540 |
| DecQN_N (N=10) | 37.9 | 616 |
| REValueD (N=10) | 45.7 | 616 |
| DAC-AC (N=10) | 64.2 | 800 |
| DAC-DDPG (N=10) | 50.6 | 576 |

$^\star$1 epoch = 10,000 grad steps

Table 3: Computational cost (Humanoid)

| Method | Runtime (s/epoch) | GPU Mem. (MiB) |
|---|---|---|
| TD3 (N=2) | 60.1 | 542 |
| SAC (N=2) | 82.5 | 549 |
| DecQN_N (N=10) | 53 | 722 |
| REValueD (N=10) | 65 | 722 |
| DAC-AC (N=10) | 133.8 | 1206 |
| DAC-DDPG (N=10) | 61 | 594 |

**Software**    We use the following software versions

- Python 3.10.0
- Pytorch 2.3.0+cu121
- Gym 0.26.2
- dm_control

## B.2 BASELINES

### B.2.1 ONLINE

For **SAC** we use automatically tuned entropy and tune the actor and critic learning rates to $7 \times 10^{-5}$ to prevent instability in high-dimensional environments. Despite this, for both Humanoid and Dog environments, learning remained unstable. For **TD3** we found the default learning rates were sufficient for all environments. Unlike the other algorithms, both **TD3** and **SAC** are most optimal online when ensemble size is small; as a result, we use the default ensemble size of $N = 2$ to ensure efficient learning. For **REValueD** we use the default architecture and hyperparameters as specified by the original authors to recreate their respective results online. For **DecQN_N**, we use the same architecture and hyperparameters as **REValueD** but set the regularisation loss coefficient $\beta$ to 0. In addition to these baselines, we also attempted to compare our method to the **Unimodal**[1] algorithm published at IEEE in 2024. Unfortunately, the available code depends on Tensorflow 1 and deprecated libraries, which we were unable to run despite multiple attempts. We also contacted the authors for support, but did not receive a response.

### B.2.2 OFFLINE

For the offline baselines, we directly report the results that were recorded by the original authors. Performance is reported using normalised returns, where a score of 0 corresponds to a random policy and 100 to an expert policy. We use these datasets to demonstrate that our adapted algorithms can integrate existing offline RL techniques to effectively mitigate distributional shift.

## B.3 HYPERPARAMETERS

Table 4: List of hyperparameters used.

|  | Hyperparameter | Value |
| --- | --- | --- |
|  | Optimiser | Adam |
|  | Critic learning rate | 1e-4 |
|  | Actor learning rate | 1e-4 |
|  | Replay size | 250000 |
|  | Discount $\gamma$ | 0.99 |
|  | n-step returns | 3 |
|  | Batch size | 256 |
|  | Target update rate | 1e-3 |
| General | Policy update frequency | 2 |
|  | Critic hidden layers | 3 |
|  | Actor hidden layers | 3 |
|  | Critic hidden dim | 256 |
|  | Actor hidden dim | 256 |
|  | Critic activation function | ReLU |
|  | Actor activation function | ReLU |
|  | Critic ensemble size | 10 |
|  | Minimum exploration, $\epsilon$ | 0.05 |
|  | $\epsilon$ Decay rate | 0.99995 |
| DAC-AC | softmax $\tau$ | 10 |

We list all hyperparameters used for our centralised critic architecture and actor networks online in Table 4 and specify the value for the regularisation hyperparameter used offline in Table 5. We seek to limit the amount of different values used for $\alpha$ to highlight that it is robust to both dataset quality and task. For DAC-DDPG-BC we found very little impact on performance when comparing hyperparameters values from $(0.5, 0.2, 0.1, 0.07)$ but found using $\alpha = 0.07$ had slightly more consistent performance across seeds. For DAC-AC-BC we found a little more variance across different hyperparameter values but still aimed to keep changes as minimal across different dataset qualities for a given task.

---

[1]Discretizing Continuous Action Space with Unimodal Probability Distributions for On-Policy Reinforcement Learning

Table 5: Regularisation hyperparameter value used offline.

| Dataset | DAC-DDPG-BC | DAC-AC-BC |
|---|---|---|
| Cheetah-Run | | |
| -random-medium-expert | 0.07 | 0.5 |
| -medium | 0.07 | 0.5 |
| -medium-expert | 0.07 | 0.5 |
| -expert | 0.07 | 0.2 |
| Quadruped-Walk | | |
| -random-medium-expert | 0.07 | 0.5 |
| -medium | 0.07 | 0.5 |
| -medium-expert | 0.07 | 0.5 |
| -expert | 0.07 | 0.5 |
| Humanoid-Stand | | |
| -random-medium-expert | 0.07 | 0.2 |
| -medium | 0.07 | 0.5 |
| -medium-expert | 0.07 | 0.5 |
| -expert | 0.07 | 0.2 |
| Dog-Trot | | |
| -random-medium-expert | 0.07 | 0.1 |
| -medium | 0.07 | 0.5 |
| -medium-expert | 0.07 | 0.1 |
| -expert | 0.07 | 0.1 |

Table 6: Details of the state and action dimension for each environment used from DM Control Suite.

| Task | Dim(S) | Dim(A) |
|---|---|---|
| Finger Spin | 9 | 2 |
| Fish Swim | 24 | 5 |
| Cheetah Run | 17 | 6 |
| Walker Walk/Run | 24 | 6 |
| Quadruped Walk/Run | 78 | 12 |
| Humanoid Stand/Walk/Run | 67 | 21 |
| Dog Walk/Trot/Run | 223 | 38 |

## C ABLATION EXPERIMENTS

We conduct an additional ablation study investigating the impact of varying the updating frequency of the actor in DAC-AC. From Figure 6 we can see that reducing the frequency of actor updates substantially reduces policy performance.

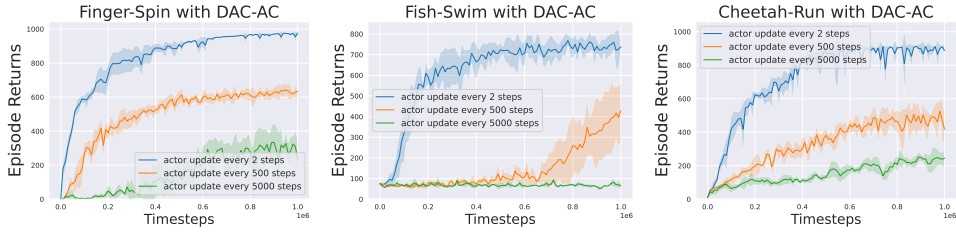

Figure 6: Performance of DAC-AC with varying actor update frequency. Solid lines show the mean over 5 seeds, and shaded regions indicate ± standard deviation. Returns are evaluated every 10,000 timesteps and averaged over 10 episodes.

## D LLM USAGE

We use LLMs in our work solely for grammatical purposes.

