# OpenReview forum: "Action Dimension Coordination via Centralised Critics for Continuous Control"
_ICLR.cc/2026/Conference — Submitted to ICLR 2026_

### Official Review · Reviewer_pkkm · 2025-10-26

**Soundness:** 3
**Presentation:** 3
**Contribution:** 2
**Rating:** 2
**Confidence:** 5

**Summary:**

This paper aims to leverage action decomposition to address the coordination problem of action dimensions in high-dimensional continuous control tasks. The authors note that existing action space decomposition methods (such as DecQN) assume independence of action dimensions, which can lead to suboptimal policies in tasks requiring fine-grained coordination. To address this, this paper proposes applying the CTDE paradigm from MARL to single-agent control. During training, a centralized critic, $Q^i$, conditions actions $a_{C_i}$ in other dimensions to learn inter-dimensional dependencies and achieve coordination. At execution, a single, decentralized actor, $\pi_\phi(a|s)$, is responsible for outputting the complete joint action.

Based on this framework, the authors instantiate two algorithms: DAC-AC and DAC-DDPG. Experiments on DeepMind Control Suite tasks demonstrate that this approach outperforms DecQN, REValueD, SAC, and TD3 in an online setting. In addition, the authors also demonstrated the policy-based regularization expansion algorithm and experimental results of this framework in the offline setting, and compared the effects of layer normalization and context Ci in the ablation experiment.

**Strengths:**

1. The core contribution of this paper is to analogize the "action dimension" to the "MARL agent," highlighting the lack of coordination limitations of existing VDN-style action decomposition methods (such as DecQN). It also introduces a mature and widely used CTDE framework from the MARL field.

2. The actor-critic structure used in this framework makes it easily integrated with existing offline RL techniques, particularly policy-constrained methods. This extends the limitations of DecQN's value-based learning and facilitates the introduction of policy regularization constraints. The state-of-the-art results achieved by DAC-AC-BC on offline benchmarks demonstrate the effectiveness of this extension.

3. The experimental results in this paper are self-contained. On complex high-dimensional control tasks such as dog, fish, and humanoid, the proposed DAC-AC and DAC-DDPG significantly outperform the basic SAC and TD3 algorithms in learning speed and variance. They also demonstrate advantages over DecQN in offline scenarios.

**Weaknesses:**

1. In the introduction, this paper contrasts "value decomposition methods" with "CTDE frameworks." This formulation is nonstandard in the MARL field and can be misleading. Value decomposition methods (e.g., VDN, QMIX, QPLEX, and QTRAN) are themselves a type of CTDE. I understand the author's intention to contrast "hypothetically independent CTDEs (e.g., VDN-style DecQN)" with "explicitly coordinated centralized critic algorithms (e.g., MADDPG)," but the current formulation is inaccurate.

2. The contribution of this paper's approach is limited. While the proposed contextual Q-value framework mitigates the VDN-style limitations of DecQN, it still lags far behind the state-of-the-art in MARL. The proposed CTDE method is essentially equivalent to a simplified version of a fully cooperative MARL algorithm, where local observations are reduced to a shared global state and the policy network is a joint action policy. Decoupled Q networks are almost MADDPG-style centralized critics, where the context Ci is set to the joint action of all other agents. In this scenario, the difficulty of learning the critic is no different from directly learning a centralized critic, completely failing to demonstrate the necessity of decomposing the action dimension. The discretization of continuous actions is also puzzling, given the numerous mature policy gradient methods in the MARL community, such as FACMAC, that support this setting.

3. The baselines considered in this paper are not comprehensive, and the experimental setup is relatively simple, making it difficult to clearly support the core argument of this paper. The online comparison only considers the classic TD3 and SAC algorithms, severely lacking a comparison with the latest performance in the field. The offline setting only considers DecQN and BC, lacking classic offline RL benchmarks such as BCQ, IQL, and TD3+BC. Furthermore, only DAC-AC-BC performs well, while DAC-DDPG-BC performs very poorly. The authors attribute this to biases in Gumbel-Softmax, but this still means that a key instance of the framework is ineffective in offline scenarios.

**Questions:**

Q1: Regarding the stability of DAC-AC policy updates: DAC-AC's actor update mechanism appears to be potentially unstable. The mechanism first involves the actor $\pi_\phi$ sampling a joint action $a \sim \pi_\phi(\cdot|s)$; then, using the sampled action $a_{-i}$ as context, it computes a new "optimal" label $a_i^{opt}$; finally, $\pi_\phi$ is trained to imitate $a^{opt}$. Consider a simple XOR game where the optimal actions are (0,0) and (1,1). Could this update strategy lead to catastrophic oscillations? For example, the actor outputs (0,0), and the critic computes the optimal label (1,1) based on the context of (0,0); the actor is updated to output (1,1); and in the next step, the critic computes the optimal label (0,0) based on (1,1). Would this unstable feedback loop prevent the policy from converging to an optimal solution (e.g., (0,1) or (1,0))? Can you design a matrix game experiment of this type to verify and illustrate this?

Q2: Regarding the trade-off between coordination and cost: The ablation experiment in Figure 4 shows that on the Dog-Walk task, using a context with 15 dimensions (|\mathcal{C}_i|=15$) yields similar performance to using 30 dimensions (fully centralized). However, I observe that using 30 dimensions consistently outperforms 15 dimensions on all the tasks shown in the figure, indicating that centralized learning is sufficient to achieve high performance, and reducing the number of observation dimensions only degrades performance. Can you provide a more detailed performance comparison, such as a scaling curve from an empty context Ci to the full-dimensional context?

Q3: As well as the issues mentioned in the weakness section.

---

> ### Author Response · Authors · 2025-11-25
> **Rebuttal Part one**
>
> We thank the reviewer for their thoughtful and detailed assessment of our work. We genuinely appreciate the suggestions and thought problems you raise that we believe will strengthen our work going forward and happily welcome any further suggestions you feel would help better motivate and strengthen our work.
>
> **W1** We thank you for raising this point. We agree that our phrasing may be misleading. As you point out our intention was to contrast value decomposition approaches where each agent's value function depends on its own observation and action with explicit centralised coordination methods such as MAAC and MADDPG. We will make this distinction explicit going forward.
>
> **W2**
>
> - **Motivation** Our goal in this work is to introduce a general framework for using centralised critics with value-decomposition style methods such as DecQN/REValueD, in order to improve coordination across action dimensions without reverting to a single monolithic critic. While our contextual critics do have access to the joint action during training, they remain factorised across dimensions, so each head only needs to reasons about a low-dimensional slice of the joint action. This is different from directly learning a fully centralised critic and, in practice, leads to improved sample-efficiency and asymptotic performance over the VDN-style DecQN baseline as shown in Figure 2. Our contribution is therefore not to propose a SOTA MARL algorithm, but to study how centralised information can be exploited in the single-agent within factored value-based methods.
>
> - **Reason for discretisation**  We deliberately focus on discrete action spaces because one of the main advantages of value-based methods such as DQN is that they estimate a value for each joint discrete action, which corresponds to a flexible categorical distribution over actions and naturally supports multi-modal policies. In contrast, common continuous-control actor critic methods (SAC/TD3 and by extension FACMAC) typically assume unimodal Gaussian or deterministic actors, partly to keep optimisation tractable; this can be restrictive when the optimal behaviour is multi-modal or highly discontinuous. One such example for instance being the XOR game you highlight in one of your later questions.
>
>      Recent work has shown that discretising continuous actions and applying value-based methods can be very effective in such settings [1]. [1] also highlight that vanilla DQN scales poorly with action dimension and therefore introduce factored utilities to trade off coordination capacity against computational efficiency. Our work is complementary, we build on this discretisation perspective and show how centralised contextual critics over factored utilities can further improve coordination across action dimensions.
>
> [1] Seyde, Tim, et al. "Solving Continuous Control via Q-learning." The Eleventh International Conference on Learning Representations. 2023.
>
> **W3**  Our focus is on high-dimensional single-agent control with factorised actions. In the online setting we include strong continuous-control baselines (TD3 and SAC), to show that, while they perform well in relatively low-dimensional domains, their performance degrades as the action dimension grows. Our main aim is then to demonstrate that centralisation within a factored value-based framework can improve upon DecQN and REValueD in this high-dimensional regime.
>
> For the offline setting, the datasets [3] utilised are specifically focused for algorithms that exploit the factorising of actions and are generated using DecQN that learn discrete actions. This makes standard continuous-action offline algorithms such as BCQ, IQL and TD3+BC non-trivial to apply without modification of their architectures. Instead we compare to the offline methods introduced in [3] as benchmarks. We will make this clarification more explicit in updated revisions of this work.
>
> Regarding DAC-DDPG-BC, it is only one simple instantiation of our framework. We see its poor offline performance as an informative negative result, indicating that naive adaption of deterministic policy methods is not well-suited here. Crucially, DAC-AC, another instantiation of the same framework, performs strongly and serves as a competitive benchmark for offline algorithms within this discretised factorised-action setting. We will make these points explicit in future iterations of our work
>
> [3] Beeson, Alex, David Ireland, and Giovanni Montana. "An Investigation of Offline Reinforcement Learning in Factorisable Action Spaces." Transactions on Machine Learning Research. 2024

---

> ### Author Response · Authors · 2025-11-25
> **Rebuttal part 2**
>
> **Q1** We thank you for raising this important point and agree it would provide additional clarity into our method. The XOR game you describe highlights a general difficult of learning symmetric multi-modal behaviour. In fact, linearly factorised value functions such as those used in VDN/QMIX and REValueD/DecQN cannon in general represent the correct value function for XOR-type problems under symmetric rewards, as a result of improper credit assignment despite their computational advantages.
>
> In DAC-AC, by conditioning each per-dimension critic on $a_{-i}$ we are in principle able to identify an optimal action $a_i^{opt}$ for each dimension, so the representation issue is alleviated. However, as you correctly point out, our current supervised imitation update, where the actror directly imitates the critic's optimal label, can in a perfectly symmetric XOR game, induce oscillatory behaviour if we relabel the entire joint action at once.
>
> A simple way tomake the policy improvement step more conservative is to "repair" only dimension at a time keeping the context fixed. Concretely given a sampled joint action $a^{samp}$, we compute $a_i^{opt}=\arg\max Q^i(s,a_{-i}^{samp})[a_i]$ and form the repaired joint action $(a_i^{opt},a_{-i}^{samp})$. The actor is then updated using one of these $n$ repaired actions rather than jumping directly to the fully greedy joint action. This mitigates the oscillation you describe while preserving the benefit of centralised criics. We will clarify this point in the revision and discuss such conservative variants of the policy updae as a practical design choice.
>
> Finally we note that related issues arise more broadly in actor-critic methods that cannot capture multi-modal solutions. For example, extending the XOR game to a continuous setting with two separated high-value regions would cause standard methods such as SAC or TD3 to average between modes. Thus the phenomenon highlighted by you example is not unique to DAC-AC but reflects a more general challenge of leaning in symmetric multi-modal tasks.
>
> This further highlights the benefits of utilising value based methods and that through careful update of the actor, our framework provides one way of systematically addressing this challenge via centralised per-dimension critics.
>
> **Q2**  We agree a scaling curve would make the coordination-cost trade-off clearer. Going forward we will revise this section to include a sweep over context sizes $|C_i|$ for each of the environments in the ablation study, to provide a more granular performance and cost-trade-off.

---

### Official Review · Reviewer_fsxY · 2025-10-27

**Soundness:** 1
**Presentation:** 2
**Contribution:** 1
**Rating:** 2
**Confidence:** 4

**Summary:**

This work re-formulates standard MDPs as a multi-agent problem under CTDE. Each action dimension is treated as an agent in a cooperative multi-agent system. Based on this framework, they introduce extensions to DDPG and actor critic, called DAC-AC and DAC-DDPG, which use a decentralized critic.

**Strengths:**

The connection between Factored Action MDPs and MARL is interesting, although the paper does not fully explore this connection in a principled way. Code is also provided in the supplementary material.

**Weaknesses:**

The fundamental motivation for this work is extremely confusing. In MARL, CTDE is a popular choice because the environment restricts us to learn a fully factorized policy where agents act independently. Thus, the goal is to maximize return over a factorized policy space. In single-agent RL, we do not have such restrictions and the policy is a joint policy defined on the joint action space, which can be represented by either a single policy or an autoregressive one. However, if we restrict the policy space to be factorized, then we are unnecessarily reducing the policy space, and making learning harder.

Furthermore, there are no theoretical guarantees or even any justifications for why the decomposition is sound. For instance the target in Algorithm 1 is not justified and appears out of nowhere. Value decomposition is also quite restrictive in MARL. To make the connections between single-agent and multi-agent setting in a more principled way, I would suggest focusing on the FA-MDP and MMDP (CTCE) and trying to find deeper connections by deriving either policy iteration or value iteration first.

The experimental results are also mixed, and certainly not convincing enough to justify the lack of theoretical results.

**Questions:**

1. How do you specify the context $a_{C_i}$?
2. Why is the policy not conditioned on the context?
3. In lines 168-169, “the decomposed functions are referred to as critics”. Why is that the case? Is the decomposed Q-value an actual Q-value, i.e. does it have an interpretation as expected cumulative return, and does it satisfy any Bellman equations?

---

> ### Author Response · Authors · 2025-11-25
> **Rebuttal**
>
> We thank the reviewer for taking the time to review our submission and for pointing out the valuable perspective of linking factored-action MDPs with MARL. We hope we can address your concerns with our responses.
>
> **W**
>
> - **Motivation**  We apologise that our motivation was not sufficiently clear. Our work is inspired by Seyed et al (2023) [1], who show that importing value decomposition ideas from multi-agent RL into the single-agent setting allows DQN to scale much better as the action dimension increases, by factorising the joint Q-function into utilities per action dimension. This factorisation stabilises learning in high-dimensional action spaces, but as you note and as also observed in Seyed et al, it sacrifices the ability to coordinate across dimensions. Our contribution is to show that we can retain the scalability benefits of this factorised representation while re-introducing coordination by using a centralised critic per actor dimension, instead of an independent utility as in [1].
>
> - **Decomposition and theoretical guarantees** Our setting can be viewed as operating on a factorised action MDP (FA-MDP), where each component of the joint action $a$ is a factor. Our primary focus is to build on the work of [1] to show how explicit coordination can enhance performance during learning. We appreciate your suggestion and agree theoretical guarantees linking FA-MDP and MMDP's can help strengthen the justification of our work and will look to develop this going forward.
>
> - **Experimental results are mixed** While the experimental results are not uniformly superior on every tasks, they consistently show that our centralised per-dimension critics are at least comparable, and in many cases improve upon the underlying value-decomposition baselines DecQN/REvalueD. This suggests that introducing coordination has a minimal negative impact on learning and can also substantially improve performance on a wide range of tasks.
>
> [1] Seyde, Tim, et al. "Solving Continuous Control via Q-learning." The Eleventh International Conference on Learning Representations. 2023
>
> **Q1** In our framework $|\mathcal{C}_i|$ is a design choice that specifies which other action dimensions are fed into the critic for dimension $i$. In the general description we  present several possibilities. For our main experiments we use $\mathcal{C}_i = {1,\dots ,N}$ for DAC-DDPG and $\mathcal{C}_i = {1,\dots ,N}\setminus \{i\}$. In keeping with the original MARL counterparts. In the ablation study we explicitly vary $|\mathcal{C}_i|$ by sampling fixed-size subsets to study how performance changes with increased contextualisation.
>
> **Q2** The context consists of other action dimensions, at execution time these are not known until the full joint action has been sampled. We therefore follow the usual CTDE pattern and centralise only the critics while keeping execution decentralised. Note that in the context of the single agent setting this simply means using conditioning the policy only on the state. As a result, the actor can be factorised over dimensions and depends only on the state.
>
> **Q3** Yes. Each $Q(s,a_{C_i})[a_i]$ is trained as a standard TD critic for the underlying MDP. In algorithm 1 we outline for each dimension, that the target can be expressed as $$y_i = r +\gamma Q^{i}(s',a_{C_{i}}')[a_{i}]$$ and minimise the Huber loss $L(y_i-Q(s,a_{C_i})$. This is simply the bellman optimality backup applied to a value function of the form
>
> $$Q^i(s,a_{C_i})[a_i] \approx E\left[\sum\gamma^tr_t|s_0=s,a_0 = (a_i, a_{\mathcal{C_{i}}}) \right].$$ All value functions share the same scalar reward and differ only in which parts of the joint action they observe, so they are action-value functions (critics) not arbitrary utility terms combined into a global critic as in VDN/DecQN.

---

### Official Review · Reviewer_4jbT · 2025-10-31

**Soundness:** 2
**Presentation:** 2
**Contribution:** 2
**Rating:** 2
**Confidence:** 4

**Summary:**

This work proposes a method to tackle environments with large action spaces that require coordination across action dimensions. This work leverages principles from MARL value decomposition methods to improve learning in the single agent domain by factorizing the action space, following the popular centralised training with decentralised execution convention.

**Strengths:**

This paper addresses issues in single agent RL using important concepts from MARL, which is interesting. A large set of experiments is provided to show the performance of the proposed method. However, there are still some points that could be improved, please see below.

**Weaknesses:**

While this paper studies an interesting problem, there are still some points that are a bit unclear and could be improved. For example, the notation is sometimes a bit confusing and makes it difficult to follow the methodology. There are also some terms that are inconsistent from a MARL perspective. The authors could have provided more deeper details about how the proposed method is different from other methods that factorise actions spaces in single agent learning, such as the mentioned DecQN. The experiments are good, but the improvements in performance are not impressive either when compared to some of the baselines. In figure 2 it would be good to see the comparison with the baselines methods without the action decomposition too, i.e., DDPG for example.

It is unclear to me if the size of C_i is environment dependent or not; if C_i can assume any size for any environment, then it is arguable to say that "full centralisation" is being used because if there is a possibility of extending the number of dimensions indeterminedly, then it means it is always possible to make it more centralised, with more dimensions.

A set of examples for C_i were introduced in lines 175-179; however, in the experiments it seems that the full set (2nd bullet point) is the one always being used.

Some minor points:
- in line 59: missing full stop "."
- in lines 175-179: some points have "." at the end of sentence, others dont
- there are missing citations in the paper; for example QMIX (line 117)
- in line 659-660: "Unimodal algorithm published at IEEE in 2024" - is there a reference instead of writing?


Please find below some more questions that reflect my concerns.

**Questions:**

1. normally in methods such as VDN, the target to approximate is based on a $Q_{tot}$ which is a mix of the individual q-values of each agent and the target is computed against that; in algorithm 1, is the target calculated based on a mix of the action values from the indices in the set $C_i$ ? What is the meaning of $[a_i]$ int his context? is it calculating the target only for the dimension i?
2. could the authors elaborate on how their method is different from other methods that factorise the action space such as DecQN?
3. could the authors elaborate on the meaning of "joint action" (line 160) in this single-agent context? does it mean that each agent can pick more than a single action? the meaning of the action dimensions in the presented context is not straightforward for the reader without deeper effort
4. in euqation $y^n$ in line 226, is the target calculated based always on the same action "$a^{opt}$"? it is unclear whether this actions changes for each value of j; it is also unclear whether this is a mix of the action dimensions or if it corresponds only to values of $a^{opt}$
5. in the experiments with different sets of $C_i$: can $C_i$ have any size? or is it dependent on the environment? have the authors considered how the right size of $C_i$ can be calculated without intervention?
6. considering that the proposed DAC-based methods perform quite high in almost every scenario, is there a specific reason for the suboptimal performance of DAC-DDPG on Finger-Spin or Humanoid-Walk?
7. could the authors elaborate on the need for K ensemble members as shown in fig 1? i cannot also find discussions about the values used for K in the paper; this value is sometimes mistaken for N, when mentioning concepts such as number of critics, ensemble sizes K, "ensemble of N=10 critics" (line 331), etc; these concepts could be made more clear for the reader

---

> ### Author Response · Authors · 2025-11-25
> **Rebuttal part 1**
>
> We thank the reviewer for taking the time out to carefully review our work and provide constructive feedback and for highlighting the extensive experiments we use to validate our methods. We hope that we can address your concerns with our rebuttal and are grateful for points you raise that we will use to clarify and improve our work going forward.
>
> **W**
>
> - **Notation/Terminology**  We thank the reviewer for pointing out that some aspects of the notation and terminology can be improved. We will carefully revise the writing to improve consistency with standard MARL terminology and to simplify notation where possible. Our framework is closely related to factorised single-agent methods such as DecQN, in the sense that we also maintain one utility function per action dimension. A key difference, however is that prior factorised methods construct a joint value using the utility $$Q_{global}(s,a) = \sum U_i(s,a_i)$$ while we train each "utility" independently hence we refer to them as critics. DecQN is therefore recovered as a special case at the level of architecture when $\mathcal{C}_i=\varnothing$ but not in the exact aggregation rule. We will clarify this distinction going forward.
>
> - **Baselines** Regarding the baselines, Figure 2 already includes strong continuous-control baselines (TD3, SAC). TD3 is a widely used successor to DDPG with improve stability and reduced overestimation bias, so we view it as a more informative comparison than with DDPG. While the experimental results are not uniformly superior on every tasks, they consistently show that our centralised per-dimension critics are at least comparable, and in many cases improve upon the underlying value-decomposition baselines DecQN/REvalueD. This suggests that introducing coordination has a minimal negative impact on learning and can also substantially improve performance on a wide range of tasks.
>
> - **Is $|\mathcal{C}_i|$ environment dependent?**  We define "full centralisation" per environment. For a given environment with a vector $\mathbf{a} = (a_1,\dots,a_N)$, a fully centralised critic for head $i$ is one that conditions on the entire joint action, i.e $\mathcal{C}_i = \{1,\dots,N\}$. If the environment has more action dimensions, the size of this fully centralised set naturally increases. In principle, $\mathcal{C}_i$ is a design hyperparameter choice and can take different sizes of structures depending on the environment. In the main experiments we chose the fully-centralised variant for simplicity and because our ablation study (Fig 4) showed that performance improves as $|\mathcal{C}_i|$ increases and then saturates.
>
> - **Examples of $\mathcal{C}_i$.** The examples in lines 175-179 were intended to illustrate the flexibility of the framework, not to claim that all of these variants are evaluated. In the current experiments we indeed use full centralisation and study different sizes of $|\mathcal{C}_i|$ in the ablation section.
>
> - **Minor points** Thanks for pointing out the grammatical mistakes/missing citations in our work, we will revise our work to correct these issues.
>
> **Q1**
>
> **Clarifying notation $[a_i]$.**  In value based methods such as DQN and DecQN the network takes the state $s$ as input and outputs a vector of Q-values, one entry per discrete action. Writing $Q(s,a)$ is therefore a slight abuse of notation, since $a$ is not actually passed into the network, instead the chosen action \textbf{indexes} the output. For this reason we use the more precise notations $Q(s)[a]$. In DecQN there is one utility function per action dimension $U^i(s)$ and the action component $a_i$ selects the corresponding entry, written $U^i(s)[a_i]$. In our framework, each per-dimesion critic additionally conditions on a set of other action dimensions $a_{\mathcal{C}_i}$.
>
> We therefore write $$Q^i(s,a_{\mathcal{C}_i})[a_i]$$
>
> to indicate that $a_{\mathcal{C}_i}$ is an input to the critic, while $a_i$ indexes the output for dimension $i$.
>
> **Targets** Unlike VDN/DecQN we do not aggregate the per-dimension utility values into a single joint value. For each per-dimension critic we form a standard TD-target for the selected component $Q^i(s,a_{\mathcal{C}_i})[a_i]$. In VDN/DecQN the aggregation is used to implicitly learn coordination, and QMIX extends this with a learned weighted sum via a hyper-network. We experimented with adapting a QMIX style mixer to the single-agent setting, but found it unstable and generally poor in performance, mainly due to the additional hyper-network required to learn mixing weights over many actions dimensions which Seyed et al (2023) [1] also note. In contrast, our framework allows explicit coordination amongst dimensions hence why we do not need to aggregate values.
>
> [1] Seyde, Tim, et al. "Solving Continuous Control via Q-learning." The Eleventh International Conference on Learning Representations. 2023

---

> ### Author Response · Authors · 2025-11-25
> **Rebuttal Part 2**
>
> **Q2** Our framework is closely related to DecQN in that we also maintain one value function per action dimension. In DecQN, each per-dimension utility $U^i(s)$ depends on the state and the chosen action index $a_i$, written $U^i(s)[a_i]$. In our notation this corresponds to the special case $\mathcal{C}_i=\varnothing$.
>
> In addition DecQN aggregates these utilities into a single joint value via a sum over dimensions. By contrast, in our method each critic has the form $Q^i(s,a_{\mathcal{C}_i})[a_i]$ allowing each $Q^i$ to condition on a subset  of other action dimensions.
>
> We train each $Q^i$ independently using the shared reward and do not aggregate them into a single joint Q-function. Coordination is therefore handled explicitly through the conditional inputs $a_{\mathcal{C}_i}$
>
> We then combine these contextual critics with a policy that is able to generate an action for a state without knowledge of $a_{\mathcal{C}_i}$ which we refer to as a "decentralised" policy. We then show, across a range of tasks, that this framework can facilitate improved sample-efficiency and/or asymptotic performance over DecQN-style baselines while retaining the same per-dimension discretisation.
>
> **Q3** The term joint action refers to the full action vector $\mathbf{a} = (a_1,\dots,a_N)$. There is still only one agent but in both DecQN and in our more general framework, a separate utility/value is produced for each action dimension.
>
> **Q4** In the expression for $y_n$ on line 226 the target is always computed with respect to the same joint optimal action $a^{opt}$. This is joint action is fixed for the target and does not change with the index $j$. The equation on line 219/220 make this explicit, showing how the joint optimal action is used to index the individual utility functions.
>
> **Q5** $\mathcal{C}_i$ is a design hyperparameter that may depend on the environment. In Figure 4 we conduct an ablation study showing how performance of DAC-AC varies as the size of $\mathcal{C}_i$ varies. Automatically learning the "right" size of $\mathcal{C}_i$ is challenging as it adds non-trivial overhead and, if mis-specified early in training, can harm exploration by restricting coordination too aggressively. For this reason, in the current work we manually select $|\mathcal{C}_i|$ per environment and study its effect via ablations, leaving efficient data-driven selection or adaptation of $\mathcal{C}_i$ as future work.
>
> **Q6**  DAC-DDPG is included as a deterministic policy instantiation of our framework. As in the original DDPG algorithm, deterministic policies can suffer from exploration difficulties. In our case this is compounded by the bias introduced by the Gumbel-Softmax relaxation for discretised factorised actions. Together, these effects likely explain why DAC-DDPG struggles on some tasks such as Finger-Spin and Humanoid-Walk.
>
> **Q7** We apologise for the confusion. In our notation $K$ denotes the number of critics (ensemble size) for each action dimension and $N$ denotes the number of action dimensions. We use ensembles because prior work (Ireland \& Montana, 2024 [2]) provide theoretical evidence that increasing ensemble size can reduce bias in value estimates, which in turn improves sample-efficiency during learning. In all our experiments we fix the ensemble size to $K=10$. This choice is reported in Table 4 and referred to as **critic ensemble size**. We will revise our work to explicitly reference this in the main text when we discuss the number of critics.
>
> [2] Ireland, David, and Giovanni Montana. "REValueD: Regularised Ensemble Value-Decomposition for Factorisable Markov Decision Processes." The Twelfth International Conference on Learning Representations. (2024)

---

### Official Review · Reviewer_yYn4 · 2025-11-01

**Soundness:** 1
**Presentation:** 3
**Contribution:** 2
**Rating:** 2
**Confidence:** 4

**Summary:**

This paper introduces two algorithms from multi-agent reinforcement learning (MARL) to address high-dimensional control in single-agent RL, following the work of Seyde et al. (2023) and others. Specifically, it proposes algorithms based on two MARL algorithms, MAAC and MADDPG, which are considered to be better than the MARL algorithm used in Seyde et al. (VDN) in the MARL literature. Experimental results on both online and offline settings show the proposed approaches are competitive to the selected baselines.

**Strengths:**

The paper has the following strengths:
1. To my knowledge, this is the first work that adopts centralized training with decentralized execution (CTDE) methods to the single-agent RL setting. While prior works have introduced MARL algorithms to address high-dimensional control problems in the single-agent case, a centralised critic approach has not been explored, which might provide better coordination across action dimensions.
2. The paper is well-written and easy to follow.

**Weaknesses:**

The paper also has some significant weaknesses:
1. The contributions are incremental. While the paper introduces two MARL algorithms with centralised critics, in an attempt to address the lack of action dimension coordination, the paper does not properly address *why* such coordination is needed. While Seyde et al. (2023) demonstrate coordination could happen across different time steps even with a decoupled value function, it’s unclear if a centralised critic provides further useful coordination. The paper does not provide any insight other than performance differences.
2. The motivation of using decentralized policies needs further justification. While the motivation of Seyde et al. (2023) is to use decentralized value function to address the high-dimensionality issue of *value-based* methods, what’s the motivation of using decentralized policies compared to a centralized policy under the actor-critic framework? Isn’t a centralized policy arguably better?
3. The paper overclaims that incorporating centralised critics improves both sample efficiency and asymptotic performance in some online RL settings. The paper only demonstrates two *actor-critic* algorithms with centralised critics perform better than a *value-based* algorithm with a decoupled Q-network. It is unclear if the occasional performance improvement is due to the difference in the critic/value function architecture (centralised vs. decoupled) or the RL algorithm difference (actor-critic vs. value-based). The experiments in Figure 4 partly address this but it is not convincing: 1) The results are based on only three environments and three seeds, which are quite limited, 2) they do not cover the case in which the condition set size is 0 (i.e., decoupled), and more importantly, 3) the difference between curves are small with overlapping shades (confidence intervals instead of standard deviation should be used for the plots).
4. The paper overclaims that the proposed two algorithms achieve state-of-the-art performance across multiple benchmarks. 1) The paper only performs empirical investigation in one benchmark with different settings (Beeson et al., 2024). 2) The paper uses standard error with five seeds as an error measure, which is not statistically significant to make such a strong claim. 3) Only one of the proposed methods appears to be better than the baselines in a subset of settings.
5. The empirical evaluation is limited to continuous control environments that do not seem to require cross action-dimension coordination (Seyde et al., 2023).

**Questions:**

Questions that might impact the rating:
1. Why is action coordination through a centralised critic needed? Seyde et al. (2023) have shown that DecQN can achieve coordination across different time steps. The results in their paper also suggest that the DeepMind Control suite (used in this paper) does not require action coordination. Could the authors provide justification for the choice of the benchmark?
2. Could the authors provide further insights or justification on the core motivation of the paper? Why would decentralized policies be better than a centralized policy under the actor-critic framework?
3. Why does SAC have such an unstable performance in humanoid tasks? What hyperparameters are used for it?
4. Are layer normalisation and huber loss applied to SAC/TD3 baseline as well?

Other minor questions that have little impact on the rating:
1. Line 387: Why would using a straight-through (ST) estimator necessarily inflate the critic’s input dimensionality? It should be possible to treat each action dimension independently as it is done in the DAC-DDPG-BC, which uses Gumbel-Softmax.
2. The paper hypothesizes that DAC-DDPG-BC suffers from bias introduced in Gumbel-Softmax. What temperature is used in Gumbel-Softmax in DAC-DDPG? Would reducing the temperature (and thus reducing the bias) improve performance in offline settings?

---

> ### Author Response · Authors · 2025-11-25
> **Rebuttal part 1**
>
> We thank the reviewer for their careful reading and constructive feedback and for highlighting the novelty and clarity of our work. We hope that we can address your concerns with our rebuttal and are grateful for points you raise that we will use to clarify and improve our work going forward.
>
> **W1** Our main contribution is  **to introduce a general CTDE framework for factorised continuous actions** in which each action dimension is treated as a cooperative "agent" with its own critic and a tunable conditioning set $\mathcal{C}_i$. When $\mathcal{C}_i=\varnothing$ our architecture reduces to the DecQN-style per dimension utilities $U^i(s)[a_i]$.
>
> DecQN then adds an explicit aggregation per step, whereas in our framework we do not aggregate utilities into a single joint Q-value. Instead each contextual critic $Q^i(s,a_{\mathcal{C}_i})[a_i]$ is trained independently. This template therefore generalises the DecQN factorisation while changing how the utilities are combined allowing for fully-centralised and local-context critics within a single template.
>
> While Seyde et al. (2023) show that some coordination can emerge indirectly through aggregating the utility values, their formulation still assumes independence between action dimensions. This leads to a credit-assignment issue in settings where the reward depends on joint patterns of actions (XOR-like structure), because the value functions cannot express which diemnsions need to change together. Our centralised per-dimension critics address this by allowing each head to condition explicitly on the other action components via $\mathcal{C}_i$, providing explicit coordination while preserving the scalability benefits of a factorised representation. Ireland \& Montana (2024) attempt to alleviate the credit-assignment issue with a regularisation term and our empirical results indicate that explicit coordination via $\mathcal{C}_i$ has a clearer and more substantial impact than regularisation, providing conceptual insight beyond the observed performance gains.
>
>
> **W2** Our work follows the factorised-action framework of Seyde et al. (2023) and Ireland \& Montana (2024). The actor in DAC-AC and DAC-DDPG is a single global network $\pi(a|s)$ that outputs all action dimensions jointly.
>
> We use the term "decentralised" only in the MARL/CTDE sense that, at execution time, the policy depends solely on the state $s$ and not on any additional centralised information. In the single-agent setting considered here, every "agent" (action dimension) observes the same state so a decentralised policy is effectively equivalent to a centralised global policy $\pi(a|s)$. The real distinction in our work is therefore between centralised critics over factored actions versus value decomposition baselines with independent critics, rather than between centralised and decentralised policies. We will revise the paper going forward to clarify this terminology and motivation.
>
> **W3** Our aim is to show that within factorised-action methods such as DecQN, replacing the decentralised critics with centralised per-dimension critics can improve coordination and credit assignment. As such, we use DecQN and REValueD as our primary baselines and show that our framework achieves at least comparable performance on all tasks while substantially improving sample efficiency/asymptotic returns on several tasks demonstrating minimal negative impact of utilising centralised critics.
>
>
> To further motivate the factorised-action framework, we also include TD3 and SAC as strong continuous actor-critic baselines with centralised critics. On the high-dimensional tasks these methods underperform compared to our discretised actor-critic variants with centralised decomposed critics, indicating that **simply switching from value based to actor critic is not sufficient** the way coordination is implemented in the critic is crucial.
>
> Regarding Figure 4, we agree that the ablation can be improved. Going forward we will include more granular increments in condition set size, report confidence intervals rather than standard deviations and increase the number of seeds used. We also note however that per seed we average over 10 episodes so in total 30 episodes are used across three seeds.

---

> > ### Author Response · Authors · 2025-11-25
> > **Rebuttal part 2**
> >
> > **W4** The offline suite of Beeson et al (2024)  is specifically designed to compare factorised-action algorithms. Within this suite, DAC-AC-BC matches or exceeds the best DecQN-CQL/IQL baseline on all random-medium-expert and medium datasets and achieves the best performance across all dataset qualities for Cheetah-Run. We will therefore revise the wording in the paper to state that our method is able to outperform the best existing factorised-action baselines on multiple datasets in this offline suite instead of specifying multiple benchmarks.
> >
> > Regarding to DAC-DDPG-BC, we agree that its performance is weaker. We view this as an informative result, that when adapting our framework to deterministic actor-critic methods that utilise Gumbel-Softmax sampling, bias issues can lead to poor offline performance.
> >
> > For each of the 5 seeds, we average over 10 evaluation trajectories (50 rollouts in total) which is common practice RL, but going forward we will increase the number of seeds used to increase confidence.
> >
> > **W5** The DM control suite we use is widely regarded as being composed of challenging continuous control tasks precisely because they require coordination across many joints. While methods such as DecQN/REValueD are capable of achieving high performance we demonstrate that coordination amongst factorised critics can be used to improve sample efficiency and asymptotic performance with minor negative impacts. We refer the reviewer to environments: fish-swim, cheetah-run, quadruped-walk, quadruped-run, humanoid-stand/walk/run dog-walk/trot/run as instances where one or both of our methods out perform the baseline factorised-action methods.
> >
> > Furthermore, our ablation study (Figure 4) demonstrates that increasing the degree of centralisation within DAC-AC by increasing the conditioning set size $|\mathcal{C}_i|$ improves performance up to a saturation point providing additional evidence that coordination across action dimension matters in these continuous-control environments.
> >
> > **Q1** Our core motivation is that existing factorised methods such as DecQN use "decentralised" utility functions per factorised-action that ignores cross-dimensional dependencies in dynamics and reward. DecQN can exhibit some implicit coordination over time through the aggregation of utilities but this value function itself assumes independence between action dimensions at each timestep.
> >
> > In contrast our framework centralises each per-dimension critic allowing $Q^i(s,a_{\mathcal{C}_i})[a_i]$ to explicitly depend on other action components. This provides an explicit mechanism for cross-dimension coordination while retaining the scalability of a factorised representation. Empirically Figure 2-3 show that conditioning on actions improves sample efficiency on several tasks with minimal negative impact, and Figure 4 shows that increasing $|\mathcal{C}_i|$ increases the rate of learning and can improve asymptotic performance.
> >
> > The DeepMind Control Suite contains a range of control problems with varying degrees of coordination, from low-dimensional tasks such as Finger-Spin and Cheetah-Run to high-dimensional, strongly coupled systems such as Humanoid and Dog. Our results indicate that the benefit of centralised critics are seen across a wide range of these tasks either through a faster rate of learning and/or improved asymptotic performance.
> >
> > Our goal is not to claim that all tasks require strong coordination but to show that explicit coordination via centralised critic is advantageous compared to prior approaches that utilise independently factorised utility functions.
> >
> > **Q2** We would like to clarify that by decentralised policy we simply meant that the execution of the policy is not reliant on sharing information between actions, it just conditions on the state information. In that sense in the context of single agent learning decentralised and centralised policies are equivalent.
> >
> > **Q3**  Actor-critic algorithms tend to increase in instability as action dimension increases most environments the SAC algorithm is used on have action dimensionality in the range of 1-10 whilst Humanoid has an action dimensionality of 21 which results in the instability observed. To give SAC the best chance to perform well we use automatically tuned entropy as in the original paper and reduce both actor and critic learning rates to $7\times10^{-5}$ as stated in the appendix (lines 652-653). We also use clipped Q-learning to prevent overestimation bias with critic ensemble size $K=2$.

---

> ### Author Response · Authors · 2025-11-25
> **Rebuttal part 3**
>
> **Q4** For SAC and TD3 we do not add layer normalisation or change the loss function, and instead use standard implementations that match the original papers. This avoids giving these baselines architecture choices that are non-standard for continuous actor-critic methods and keeps our comparison closer to widely used SAC/TD3 baselines.
>
> We would like to emphasise that our main comparison is with algorithms that use decomposed critics. We use Huber loss and layer normalisation for the decomposed-critic family of methods following Seyde et al. (2023) and Ireland \& Montana (2024). While we do an ablation analysis studying the effects on these algorithms we report their results in Figure 1 using their standard architectures to give a fair comparison.
>
> **M1** Using a straight through estimator requires using a one hot encoding for action dimensions which scales the vector representation of action dimensions by a factor of $k$ where $k$ is the number of bins chosen.
>
> **M2** In DAC-DDPG we use the default Gumbel-Softmax temperature $\tau$. Intuitively, decreasing $\tau$ makes the samples close to one-hot and therefore reduces bias but at the cost of gradient variance and less smooth optimisation. An effect that is particularly problematic in the offline setting, where gradients must be estimated from a fixed dataset.

---

### Meta-Review · Area_Chair_szDM · 2026-01-06

**Summary:**

The contribution is very incremental and motivation is not clear.

**Reviewer Concerns:**

The notations were clarified, but the motivation was not made clear.

**Reviewer Scores:**

N/A

---

### Decision · Program_Chairs · 2026-01-26

Reject